# Hydrogen Gas Generation Using Self-Assembled Monolayers (SAMs) of 5,10,15,20-Tetrakis (*p*-Thiophenol) Porphyrin on a Gold Electrode

Ibrahim Elghamry *, Abdulrahman S. Alablan and Mamdouh E. Abdelsalam

Department of Chemistry, College of Science, King Faisal University, P.O. Box 400, Al-Ahsa 31982, Saudi Arabia;
a.alablan@hotmail.com (A.S.A.); mamdouh.abdelsalam@gmail.com (M.E.A.)
* Correspondence: ielghamry@kfu.edu.sa or elghamry@hotmail.com; Tel.: +966-595598122

**Abstract:** A novel approach was used to synthesize the 5,10,15,20-tetrakis (*p*-thiophenol) porphyrin (TPTH-P) (**2**), which involved the demethylation of tetra (*p*-anisole) porphyrin (**1**) in the presence of $ZnCl_2$ as a catalyst and DMF as a solvent at 100 °C. The demethylation step was followed by an acidification step with concentrated $H_2SO_4$ to yield the desired TPTH-P (**2**) in an almost quantitative yield (95%). The chemical structure of the synthesized porphyrin thiol (TPTH-P) (**2**) was verified through spectroscopic techniques (NMR, IR, UV-Vis). The catalytic activity of TPTH-P in the hydrogen evolution reaction (HER) was investigated in 0.1 M of $H_2SO_4$ and 1 M of $KNO_3$. A self-assembled monolayer (SAM) of TPTH-P was formed on a gold electrode. The immersion time during SAM formation and the electrochemical activation cycles in $H_2SO_4$ were found to be important to enhancing the activity of the Au-TPTH-P electrode in the HER. Contact angle measurements and electrochemical techniques, including cyclic voltammetry (CV), linear sweep voltammetry (LSV), electrochemical impedance spectroscopy (EIS), and chronoamperometry, were used to characterize and evaluate the electrochemical activities of the SAM.

**Keywords:** self-assembled monolayers (SAMs); hydrogen evolution reaction (HER); porphyrin; cyclic voltammetry (CV); linear sweep voltammetry (LSV); electrochemical impedance spectroscopy (EIS); chronoamperometry

## 1. Introduction

Self-assembled monolayers (SAMs) have attracted great attention and are used in a wide range of applications, including in molecular and electronic devices, sensors, electrocatalysis, and clinical applications, and for investigating electron flux [1–3]. SAMs can be defined as the spontaneous assembly of a molecule monolayer on a substrate surface. These molecules are chemisorbed on the surface via the formation of a stable covalent bond. Additionally, a well-ordered SAM structure is attributed to intramolecular and intermolecular interactions [4,5]. Typically, the molecules involved in a SAM can be dissected into three parts. The first part is the head group, which is a functional group that has a strong affinity for the substrate, resulting in the direct connection of the head to the substrate on the surface. Examples of head group molecules are hydroxide (OH), thiol (SH), and carboxylic acid (COOH). The second part is known as the spacer or tail and links the head group to the functional end group, such as an alkyl chain or aromatic system. The third part is the functional end group, which is the group that provides the characteristics of the surface [5]. The hydrophilicity or hydrophobicity of a surface can be amended by using an appropriate functional end group. For example, an OH group will make the surface hydrophilic, while a $CH_3$ group will make the surface hydrophobic [6]. Several metals have been used as a substrate for SAMs, including silver, copper, gold, platinum, and palladium. Nevertheless, gold substrates are commonly employed and are considered the standard for their various merits. These include the fact that gold is inert and readily forms a stable

SAM, which is known to be stable for weeks when the substrate is kept in solution [7]. Additionally, the films formed on a gold electrode are well organized when compared with those of other metals such as platinum [8].

Organic sulfur molecules, such as thiol, sulfides, and disulfides, have been widely used to modify the surface of the gold substrate via the formation of a highly stable covalent (Au-S) bond.

Porphyrin scaffold SAMs have been reported in the literature for different applications [9]. Direct anchoring of porphyrin SAMs on the surface of the substrate will provide great flexibility in engineering and designing surface-terminal functional groups. Therefore, combining the unique properties of both the gold substrate and the porphyrin molecules will enhance the applications of this class of materials in electrocatalysis [9,10]. Many methods have been used to tailor porphyrin SAMs on the substrate surface; however, immersion of the substrate in a dilute solution containing the desired molecule is commonly utilized [7,11]. The modification of a gold surface with a SAM containing sulfur has been widely studied in the literature due to the simplicity of the assembly as well as the outstanding stability of the bond formed between gold and sulfur [10–13]. Xiaoquan et al. examined the formation of a metal-free alkylthioloporphyrin SAM on a gold surface. An electrochemical investigation revealed that when the length of the bond linking alkanethiol to porphyrin increases, electron transfer from the electrode to the solution decreases and vice versa. Additionally, the effectiveness of inserting a transition metal ion into the center of the porphyrin was also examined. It was found that inserting a transition metal increases the electron flux from the electrode to the SAM, which is attributed to porphyrin structure distortion after inserting the metal [14]. In another report, a glassy carbon electrode and a gold electrode with thiol-cobaltous porphyrin SAMs were modified to detect the presence of hydrogen peroxide throughout an oxygen reduction reaction (ORR) [9]. Furthermore, an alkylthioloporphyrin–copper complex was immobilized on a gold electrode using disulfide and investigated by Federico J. Williams et al. [15]. A SAM was formed by soaking the Au substrate in a porphyrin solution for 24 h. It was found that adsorption of the alkylthioloporphyrin–copper complex on the gold surface could lead to the demetallation of a copper ion from the porphyrin scaffold [15].

Subsequently, this led to a decrease in the electron flux and the deactivation of the system [15,16].

In this work, a thiophenolporphyrin, namely, 5,10,15,20-tetra (*p*-thiophenol) porphyrin (TPTH-P) (**2**) (Scheme 1), was designed with four thiophenol terminal groups to enable the direct assembly of the porphyrin $18\pi$-electron systems on top of the Au electrode via a strong covalent S-Au bond, which subsequently facilitated electron conductivity between the Au surface and the porphyrin $18\pi$-electron system. Generally, sulfur forms a highly stable covalent bond with the gold surface; accordingly, an ultra-thin monolayer of thiol-porphyrin is formed. SAM formation on the gold electrode is confirmed via measuring the contact angle, monitoring the behavior of the electrode in a ferro/ferricyanide redox couple, and employing electrochemical impedance spectroscopy (*EIS*). Then, the gold electrode modified with the TPTH-P SAM was used as a catalyst for the HER. The effect of potential cycling, namely the activation protocol (AC), on the catalytic activity of the SAM was studied. Catalytic activity descriptors, such as the overpotential and the Tafel slope, were taken into consideration to compare the synthesized catalyst in this work with previously reported catalysts. Moreover, the effect of soaking time on the organization of the SAM was studied. Finally, the stability of the catalyst was successfully tested and monitored through chronoamperometry.

**Scheme 1.** Synthesis and characterization of 5,10,15,20-tetrakis (*p*-thiophenol) porphyrin (**2**), abbreviated to TPTH-P, from 5,10,15,20-tetrakis (*p*-thioanisole) porphyrin (**1**), abbreviated to TPT-P.

## 2. Results and Discussion

### 2.1. Organic Chemistry

Synthesis and Characterization of 5,10,15,20-Tetrakis (*p*-Thiophenol) Porphyrin (TPTH-P) (**2**)

In the literature, only one approach has been reported for the synthesis of 5,10,15,20-tetrakis (*p*-thiophenol) porphyrin (TPTH-P) (**2**) [17]. This synthesis follows a long protocol, which firstly requires the preparation of the tetrasulfonato derivatives followed by deoxygenative reduction to the thiol using a mixture of diarylsulfides/triphenylphosphine [18]. However, it suffers from many drawbacks and disadvantages like using a large excess of very expensive reagents and producing other side reduction products.

Therefore, in this work, the target free-base 5,10,15,20-tetrakis (*p*-thiophenol) porphyrin (TPTH-P) (**2**) was synthesized using a new and unique approach. This method includes demethylation of the tetra (*p*-anisole) porphyrin (TPT-P) (**1**) (Scheme 1) in the presence of $ZnCl_2$ as a catalyst and DMF as a solvent at 100 °C [16], followed by acidification with concentrated $H_2SO_4$ to yield the desired TPTH-P (**2**) in an almost quantitative yield (95%). The purity and yield of the produced thiol combined with the availability and use of cost-effective $ZnCl_2$ as a catalyst make this approach the method of choice for the synthesis of the free-base 5,10,15,20-tetrakis (*p*-thiophenol) porphyrin (TPTH-P) (**2**).

The chemical structure of the synthesized porphyrin was characterized by spectroscopic techniques (NMR, IR, UV-Vis) and confirmed by comparing these data with previously reported spectroscopic data of the same compound. Two noticeable signs in the $^1$H-NMR spectra of TPTH-P (**2**) (Figure 1A) confirmed the completion of the demethylation reaction (Scheme 1). Firstly, the absence of the methyl group signal of the starting material (Figure 1B) at δ = 2.62 ppm. Secondly, the appearance of a singlet signal at δ = 3.82 ppm, which exchanges and disappears after the addition of $D_2O$ and is attributed to the S-H bond (Figure 1C).

The UV–Vis absorption spectra of the (*p*-thiophenol) porphyrin (TPTH-P) (**2**) (Figure 2) exhibited a slight blue shift in the Soret band at λ = 421 nm compared with that of the starting (*p*-thioanisole) porphyrin (**1**), which appeared at λ = 424 nm (Figure 2).

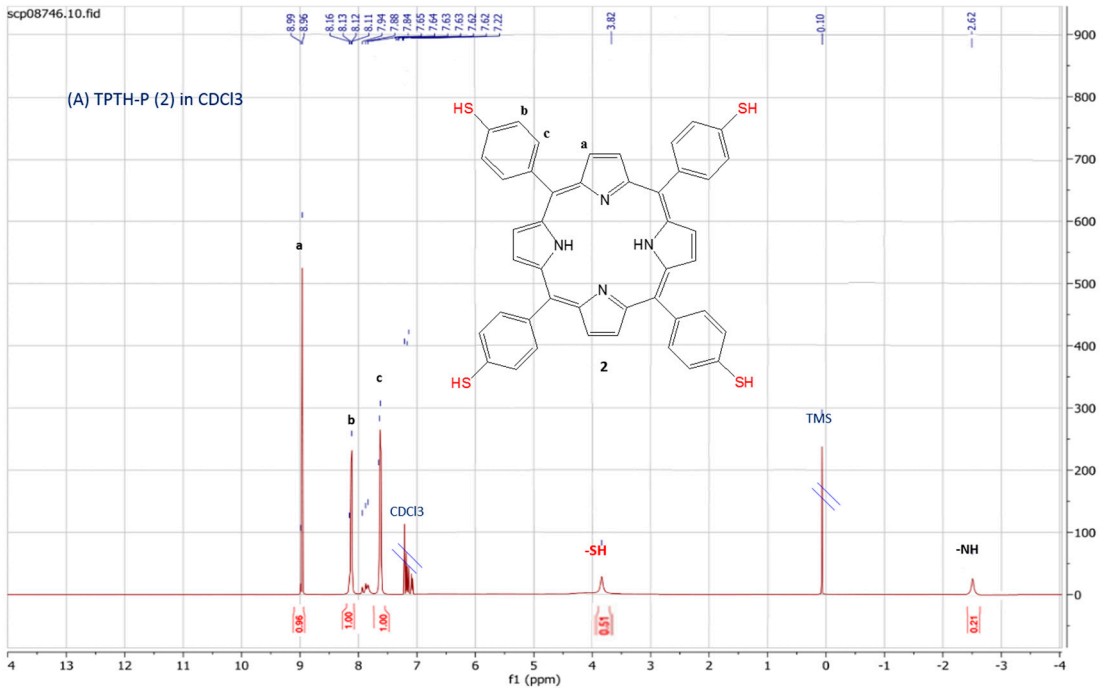

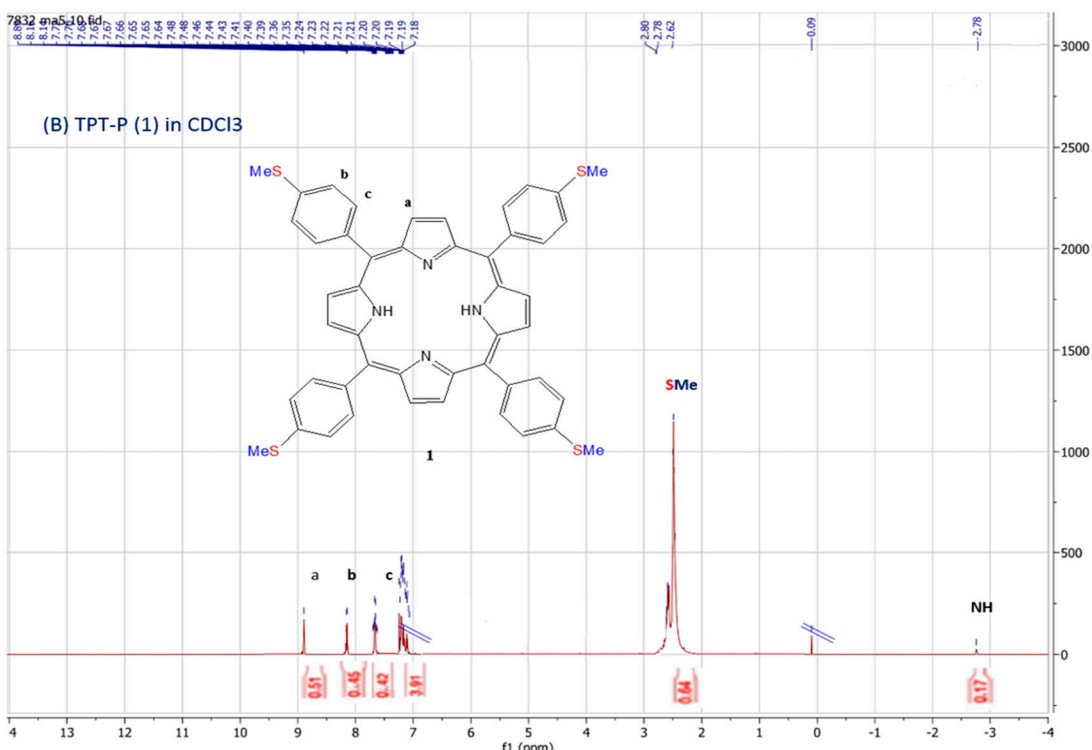

**Figure 1.** *Cont.*

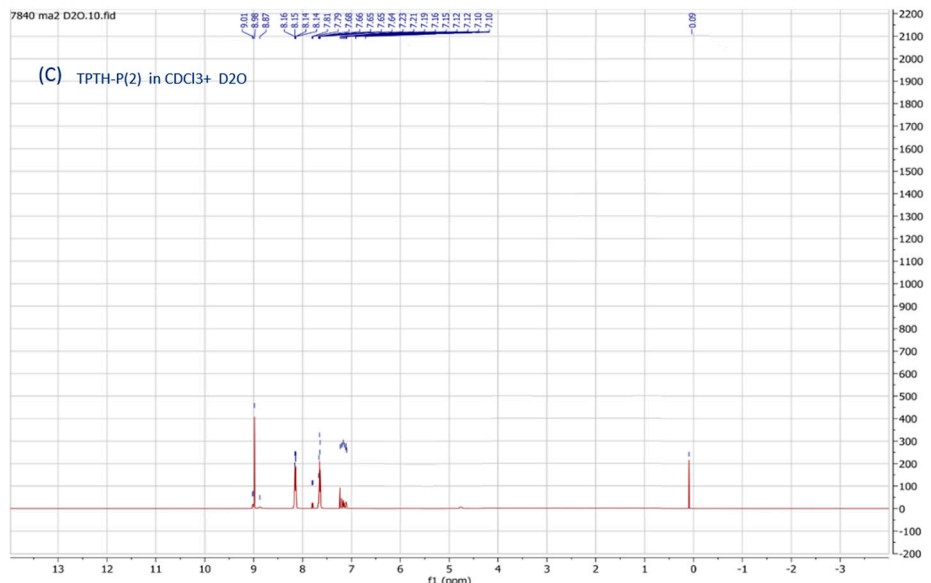

**Figure 1.** $^1$H-NMR spectra (CDCl$_3$): (**A**) tetrakis (*p*-thiophenol) porphyrin (TPTH-P) (**2**); (**B**) tetrakis (*p*-thioanisole) porphyrin (TPT-P) (**1**); (**C**) TPTH-P (**2**) + deuterated water (D$_2$O).

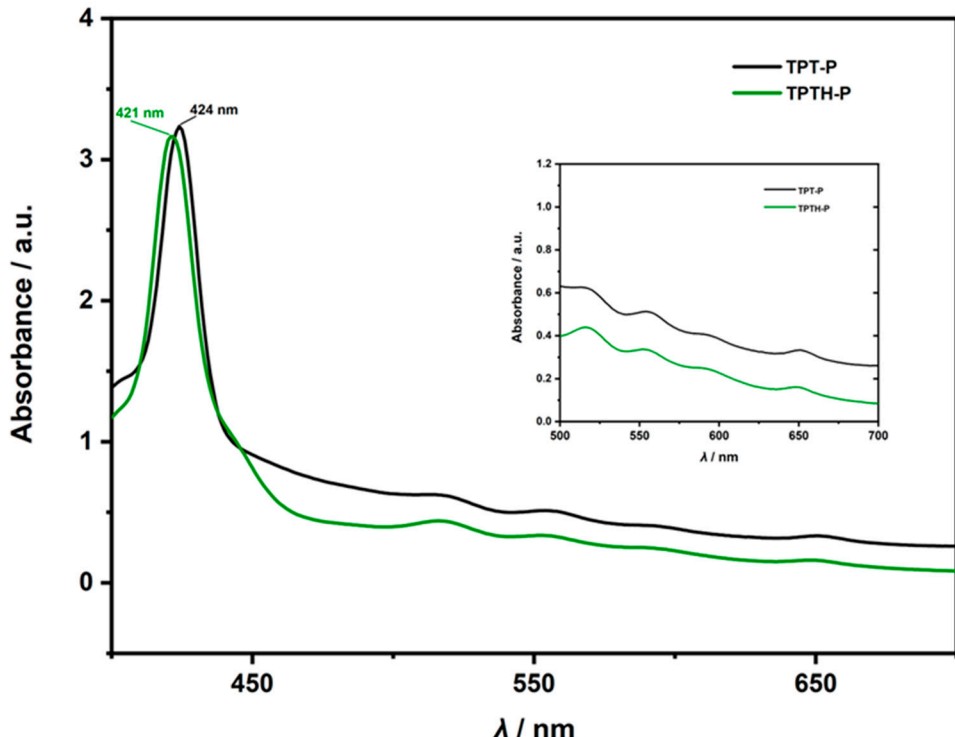

**Figure 2.** UV-Vis absorption spectra (DCM, CH$_2$Cl$_2$) of tetrakis (*p*-thiophenol) porphyrin (TPTH-P) (**2**) (green) and tetrakis (*p*-thioanisole) porphyrin (TPT-P) (**1**) (black).

The FTIR spectra of the (*p*-thiophenol) porphyrin (TPTH-P) (**2**) (Figure 3) exhibited a very sharp peak at 2960 cm$^{-1}$ attributed to a S-H group, confirming demethylation, in addition to three distinctive peaks at N-H vibrational modes of 3300, 1136, and 996 cm$^{-1}$, which are assigned to the stretching, in-plane, and out-of-plane deformations, respectively, of (*p*-thioanisole) porphyrin (**1**) (Figure 3).

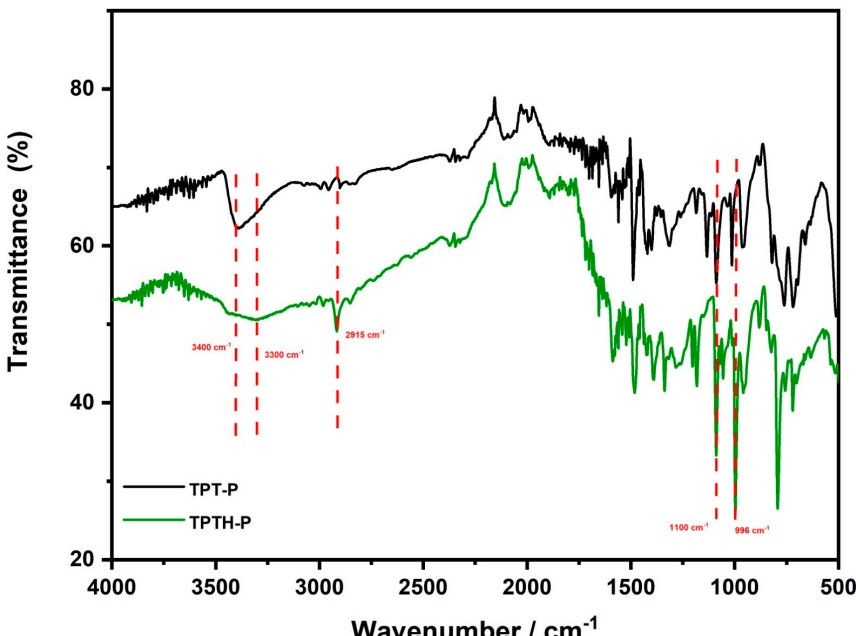

**Figure 3.** FTIR spectra of tetrakis (*p*-thiophenol) porphyrin (TPTH-P) (**2**) (green) and tetrakis (*p*-thioanisole) porphyrin (TPT-P) (**1**) (black). The red dotted lines indicate the comparison and position of the main functional groups in both porphyrins (NH, and S-H respectively).

*2.2. Electrochemistry*

2.2.1. Formation and Characterization of the SAM on the Au Surface

A TPTH-P SAM is readily formed on the gold's surface when it is soaked in a chloroform solution of TPTH-P. TPTH-P molecules anchor to the gold surface via a Au-S covalent bond [13]. The contact angle was measured to monitor the formation of the SAM. TPTH-P is an amphiphilic molecule; the porphyrin ring itself is hydrophobic, while the SH groups are hydrophilic [14]. Figure 1 shows the contact angle and shape of the water droplet on top of the Au surface before and after the formation of the SAM. It is obvious that the unmodified surface exhibited hydrophobicity toward the water droplet, resulting in a spherical shape, which is a typical phenomenon observed on clean gold surfaces [18]. On the other hand, the surface of the gold electrode after modification with thiol-porphyrin became significantly wettable (hydrophilic), and the water droplet was less spherical. The measured contact angles for the gold electrode unmodified and modified with thiol-porphyrin are ϕ = 83° and 46°, respectively (Figure 4). This result indicates that thiol-porphyrin successfully assembled and formed a SAM on the gold electrode.

Cyclic voltammetry was also used to confirm TPTH-P SAM formation on Au. Figure 5 shows the cyclic voltammograms recorded in 10 mM $K_4Fe(CN)_6$ and 1 M $KNO_3$ at a scan rate of 50 mV·s$^{-1}$. Both the unmodified and modified Au electrodes exhibit reversible behavior. The current density produced at the modified electrode is lower than the current density produced by the bare electrode. This decrease in current density could indicate the formation of thiol-porphyrin on the gold surface. It has been reported that modifying the gold surface with a SAM decreased electron tunneling from the electrode to the SAM; as a result, the produced current was reduced [14].

To further confirm the SAM formation on the gold electrode, electrochemical impedance spectroscopy (EIS) was utilized to monitor the charge transfer resistance of the Au and Au-TPTH-P electrodes. Figure 6 shows Nyquist plots for the bare Au and Au-TPTH-P electrodes. It is clear that the Au-TPTH-P electrode exhibited more resistance compared with the bare gold electrode, with Rs = 1.49 and 4.6 (Ω.cm$^2$) for the Au electrode and Au-TPTH-P electrode, respectively, confirming that the surface of the gold was successfully modified with a thin thiol-porphyrin film.

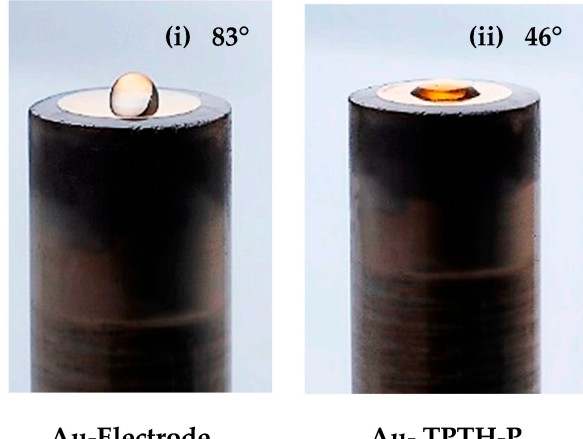

**Figure 4.** The contact angle on a bare gold electrode (**i**) and a modified gold electrode with a SAM (Au-TPTH-P) (**ii**).

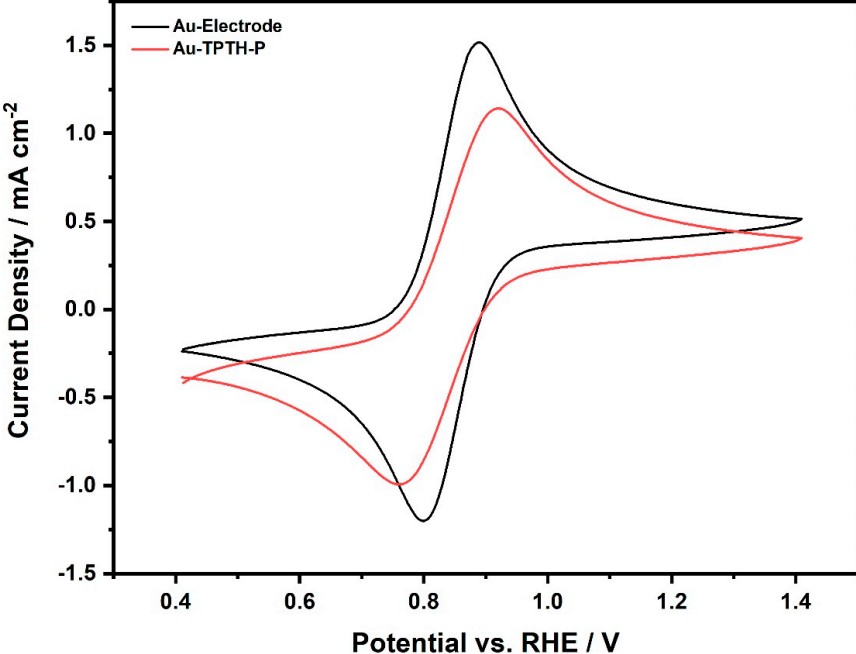

**Figure 5.** Cyclic voltammogram for the bare Au electrode (black) and the Au-TPTH-P (red) in ferricyanide 10 mM and 1 M KNO$_3$ at scan rate of 50 mV·s$^{-1}$.

Table 1 reports the contact resistance (Rs) values for the Au and Au-TPTH-P electrodes.

**Table 1.** *Rs* values for the investigated Au and Au-TPTH-P electrodes in Ar-saturated 0.1 M H$_2$SO$_4$ and 1 M KNO$_3$ solution.

|  | Unmodified Au Electrode | Au-TPTH-P |
|---|---|---|
| Rs/Ω.cm$^2$ | 1.49 | 4.6 |

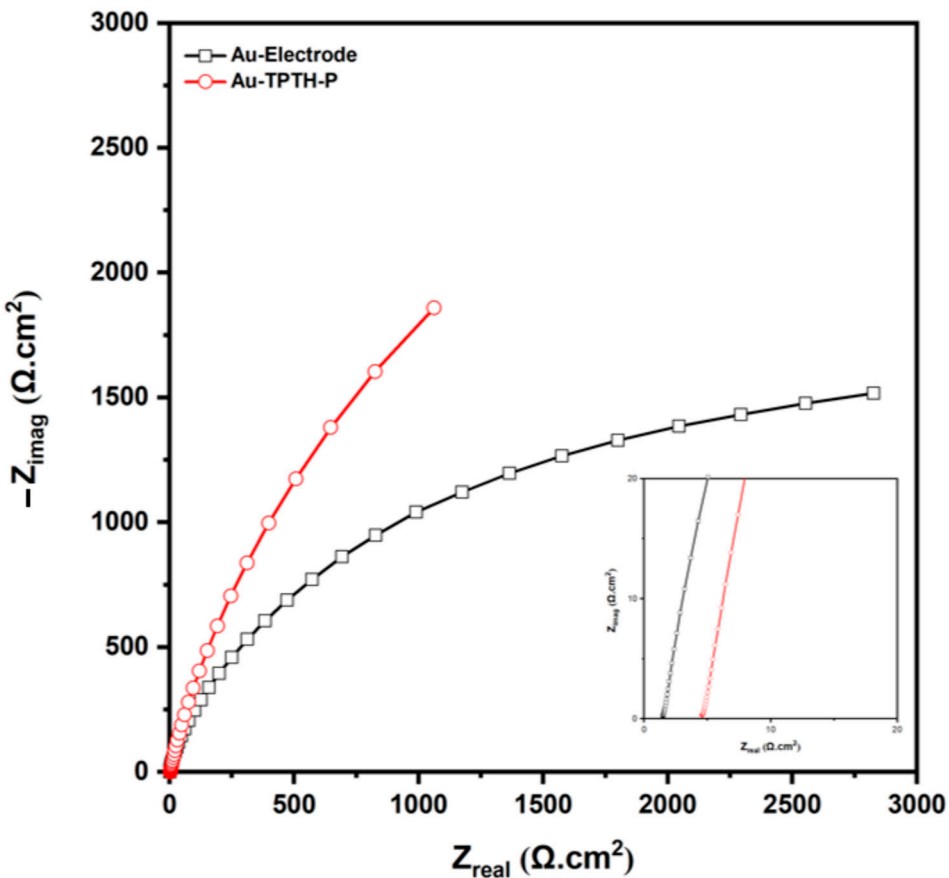

**Figure 6.** Nyquist plots of the unmodified gold electrode (black) and the modified electrode with TPTH-P (red) in 0.1 M $H_2SO_4$ and 1 M $KNO_3$; inset shows a magnified view.

2.2.2. Electrochemical Activity of the Au-TPTH-P Electrode

Hydrogen can be produced in acidic conditions through the water-splitting reaction according to Equation (1):

$$4H^+ + 4e^- \rightarrow 2H_{2\ (g)} \tag{1}$$

The electrochemical activity of the Au-TPTH-P electrode in the HER was investigated in an aqueous acidic solution containing 0.1 M $H_2SO_4$ and 1 M $KNO_3$ using linear sweep voltammetry (LSV), as shown in Figure 7. The overpotential was calculated using the following equation [19]:

$$|\eta| = E_{eq} - E_{applied} \tag{2}$$

where $\eta$ is the overpotential, $E_{eq}$ is the equilibrium potential ($E_{eq}$ = 1.23 V vs. RHE), and $E_{applied}$ is the practical potential. When comparing the bare Au electrode with the Au-TPTH-P electrode, the latter exhibited poor catalytic activity in the HER. In addition, the calculated overpotentials ($\eta$ at 3 mA·cm$^{-2}$) for the bare Au and Au-TPTH-P electrodes were 0.183 V and 0.205 V vs. RHE, respectively. This initial outcome suggests that the thiol-porphyrin SAM has poor catalytic activity in the hydrogen evolution reaction.

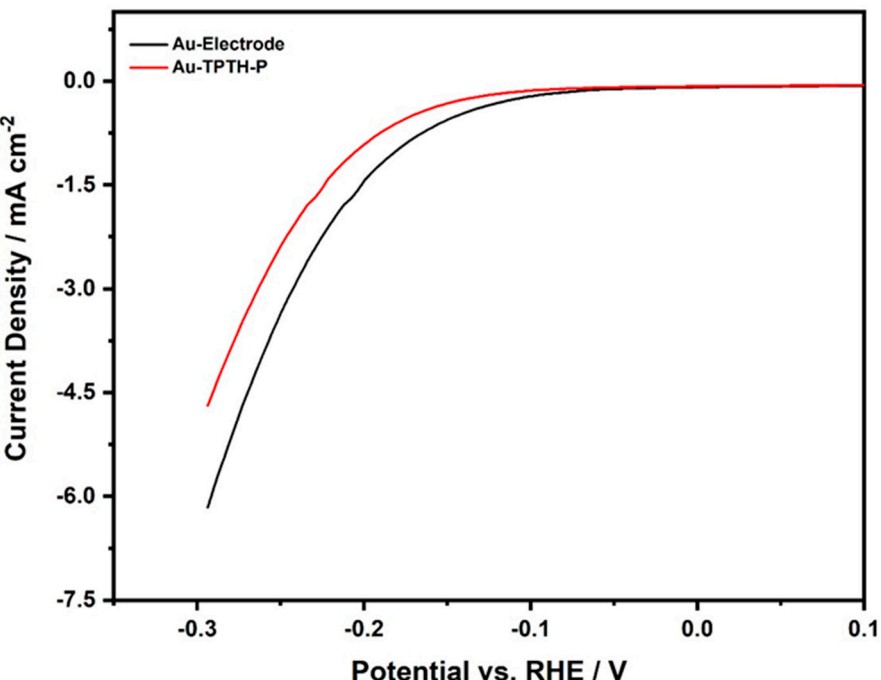

**Figure 7.** Linear sweep voltammogram of the bare gold electrode (black) and the electrode modified with TPTH-P (red) in acidic medium of 0.1 M $H_2SO_4$ and 1 M $KNO_3$ at scan rate of 10 mV·s$^{-1}$.

### 2.2.3. Electrochemical Activation of the Au-TPTH-P Electrode

The electrochemical activation process of the Au-TPTH-P electrode involves consecutive cycling in 0.1 M $H_2SO_4$ and 1 M $KNO_3$, as shown in Figure 8.

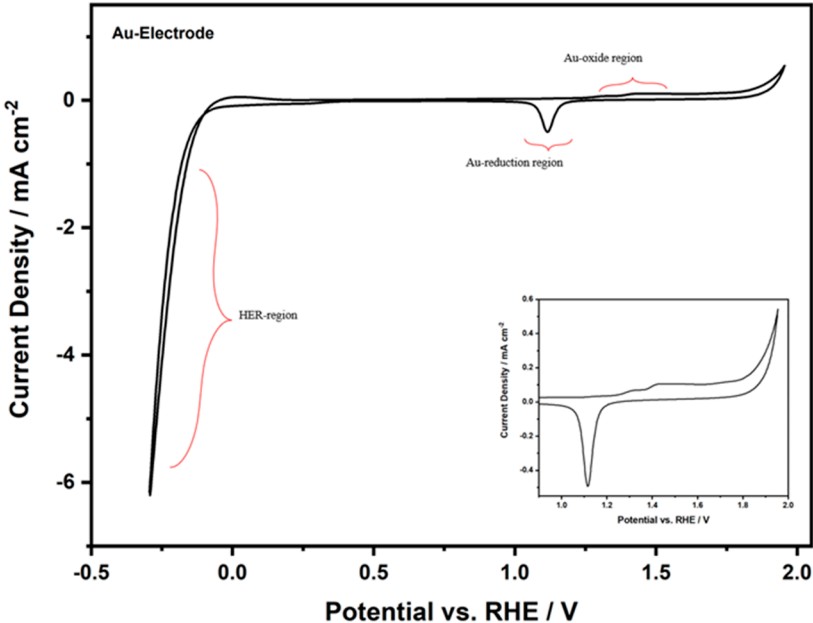

**Figure 8.** Cyclic voltammogram of the Au-TPTH-P electrode in 0.1 M $H_2SO_4$ and 1 M $KNO_3$ at scan rate of 10 mV·s$^{-1}$; inset: the redox peaks of the gold electrode.

These activation cycles were found to have a significant effect on the catalytic activity of the SAM. Figure 9 shows the linear sweep voltammogram of the Au and Au-TPTH-P electrodes in an acidic medium (0.1 M $H_2SO_4$ and 1 M $KNO_3$) at a scan rate of 10 mV/s after being subjected to the activation cycles illustrated in Figure 8. The current density (at 3 mA·cm$^{-2}$) was compared between the unmodified and modified gold electrodes. Clearly,

the consecutive potential cycling of the Au-TPTH-P electrode significantly enhanced the catalytic activity in the HER. Compared with the unmodified Au electrode, the Au-TPTH-P electrode exhibited a significant decrease in overpotential ($\eta$ at 3 mA·cm$^{-2}$), reaching 0.127 V after 10 cycles, along with an increase in the current density to about 10 mA·cm$^{-2}$. This improvement in overpotential, along with high current density, could be attributed to the change in the structural orientation of the SAM after cycling the electrode potential. This change to a highly organized structure could make the imine groups more accessible to catalyze the HER. Additionally, TPTH-P was anchored directly on the gold electrode without any spacers. Such a modification is advantageous, owing to the short distance between the Au electrode and TPTH-P, which in turn could expedite the electron tunneling through the SAM, thus reducing protons to molecular hydrogen [20–22].

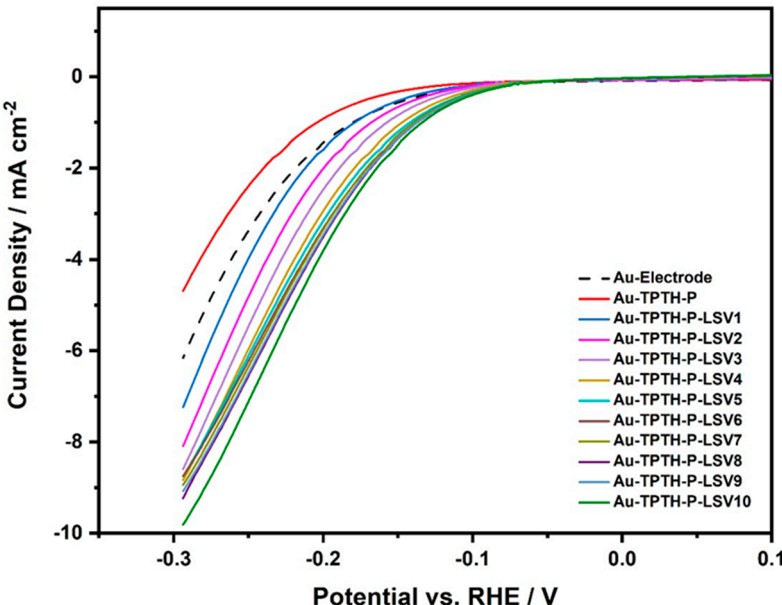

**Figure 9.** Comparative linear sweep voltammograms for Au-TPTH-P before and after activation in the acidic medium of 0.1 M H$_2$SO$_4$ and 1 M KNO$_3$ at scan rate of 10 mV·s$^{-1}$.

### 2.2.4. The Effect of Immersion Time on Catalytic Activity

The duration of soaking the gold electrode in TPTH-P solution during SAM formation was found to play a key role in influencing the quality of the formed layer and its catalytic activity in the HER. The effect of immersion time was studied by conducting LSV on the gold electrode in an acidic solution (0.1 M H$_2$SO$_4$ and 1 M KNO$_3$) after different soaking durations. The gold electrode was thoroughly polished and immersed in a solution containing TPTH-P for 2 h, 4 h, or 6 h. Figure 10 shows the linear sweep voltammograms for Au-TPTH-P immersed in a thiol-porphyrin solution for different durations. A poor catalytic activity was observed in the HER when Au-TPTH-P was soaked for 2 h. However, when the electrode was immersed for 4 h and the optimum 6 h, the performance was significantly improved, and the potential was positively shifted. This catalytic activity enhancement could be due to an increase in the SAM quality of the gold electrode, becoming more organized.

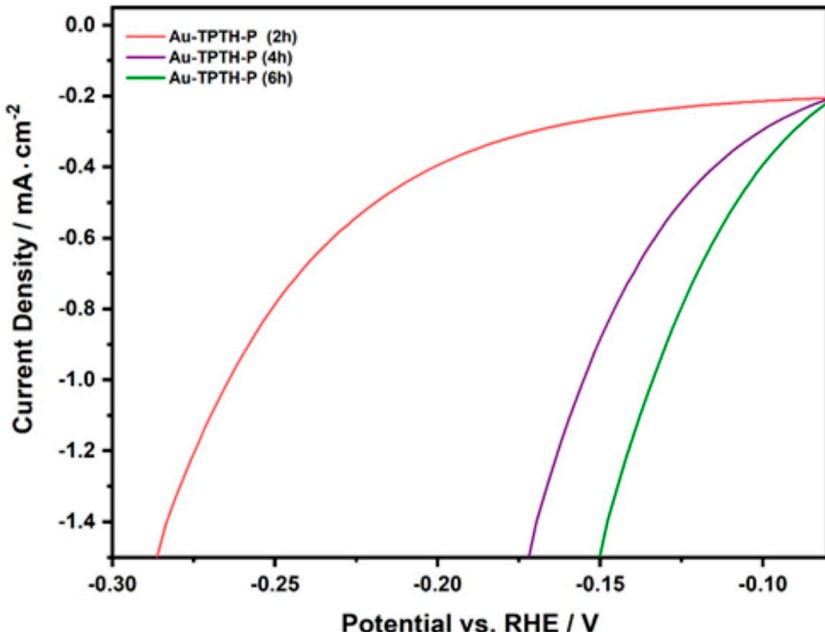

**Figure 10.** The effect of immersion time on the catalytic activity of the Au-TPTH-P electrode in acidic media (0.1 M $H_2SO_4$ and 1 M $KNO_3$), at scan rate of 10 mV/s$^{-1}$.

To further understand the HER kinetics, Tafel plots for the bare Au and Au-TPTH-P electrodes were calculated from the curves in Figure 10. The plot relates the current density to the overpotential according to Equation (3):

$$\log(i) = \log(i_0) + \frac{\eta}{b} \tag{3}$$

where the current density and the exchange current density are denoted as $i$ and $i_0$, respectively, and b represents the Tafel slope, while $\eta$ is the overpotential. A small Tafel slope value implies that the current density increases with a slight change in overpotential, which indicates a faster reaction rate. Hence, Tafel slopes were used as an indicator of the catalytic activity.

The HER mechanism in acidic conditions follows either the Volmer–Heyrovsky mechanism or the Volmer–Tafel mechanism [23].

Tafel slopes can help indicate the appropriate mechanism of reaction. In the HER, the rate-determining step could be:

(i)     The adsorption of a proton on the catalytically active site (Tafel slope of 120 mV·dec$^{-1}$).
(ii)    The electrochemical desorption of molecular hydrogen (Tafel slope ranging from 40 to 120 mV·dec$^{-1}$).
(iii)   The chemical desorption of molecular hydrogen (Tafel slope ranging from 30 to 40 mV·dec$^{-1}$).

Steps (ii) and (iii) are the so-called Volmer–Heyrovsky and Volmer–Tafel mechanisms, respectively. However, these Tafel slope values are liable to change depending on the nature of the catalyst. Figure 11 shows the Tafel plots for the Au and Au-TPTH-P electrodes before activation and the Au-TPTH-P electrode after activation. Furthermore, Table 2 reports the key performance parameters to compare the performance of the SAM reported in this work with other catalysts reported in the literature. The Tafel slope for the Au-TPTH-P electrode was 112 mV·dec$^{-1}$ after one activation cycle, which drastically decreased to 81 mV·dec$^{-1}$ after ten activation cycles. This is presumably due to the self-organization of the SAM on the gold electrode, as the subsequently well-organized structure after potential cycling makes the catalytically active site more accessible.

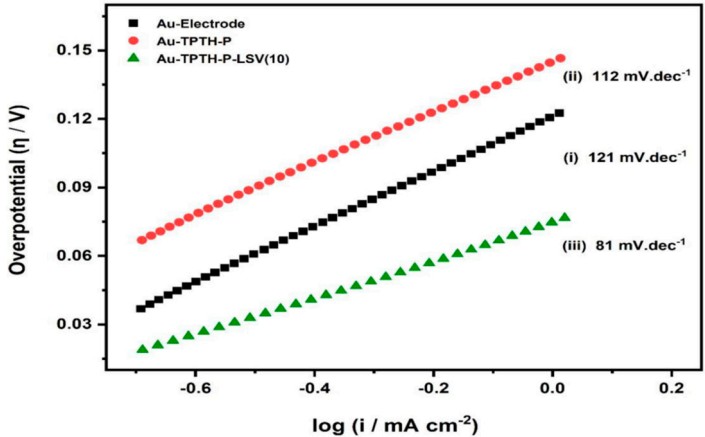

**Figure 11.** Tafel slopes for the bare gold electrode (**i**), the Au-TPTH-P electrode before activation (**ii**), and the Au-TPTH-P electrode after activation (**iii**) in acidic conditions of 0.1 M H$_2$SO$_4$ and 1 M KNO$_3$.

**Table 2.** The overpotential of previously reported catalysts compared with the SAM in this work.

| Catalyst | | | | |
|---|---|---|---|---|
| **MeOTTP** | **Environment** | **η at 3 mA·cm$^{-2}$** | **Tafel Slope (b)** | **Ref** |
| [ERGO@CoTMPyP] | pH 2.0/0.1 M NaH$_2$PO$_4$ | 562 mV | 111 mV·dec$^{-1}$ | [24] |
| MoS$_2$ | 0.5 M H$_2$SO$_4$ | 530 mV | NA | [25] |
| Pt/C | 0.5 M H$_2$SO$_4$ | 182 mV | 50 mV·dec$^{-1}$ | [26] |
| TPTH-P | 0.5 M H$_2$SO$_4$ | 15.5 mV | 30 mV·dec$^{-1}$ | [27] |
| * It is expected that increasing the H$_2$SO$_4$ concentration to 0.5 M would decrease both the b and η values. | 0.1* M H$_2$SO$_4$ | 127 mV | 81 mV·dec$^{-1}$ | This work |

### 2.2.5. Chronoamperometry

The stability of the Au-TPTH-P catalyst was investigated in acidic media (0.1 M H$_2$SO$_4$ and 1 M KNO$_3$) via chronoamperometry. The SAM-modified electrode was stepped from a potential where a non-faradic current occurs to potentials where a faradic current exists. Figure 12 represents the stepped potential of the working electrode at various potentials (−0.184, −0.224, and −0.274 V vs. RHE). The TPTH-P-modified electrode was active for up to one hour. These results imply that the thiol-porphyrin SAM is a promising catalyst for the HER, with high stability under harsh conditions.

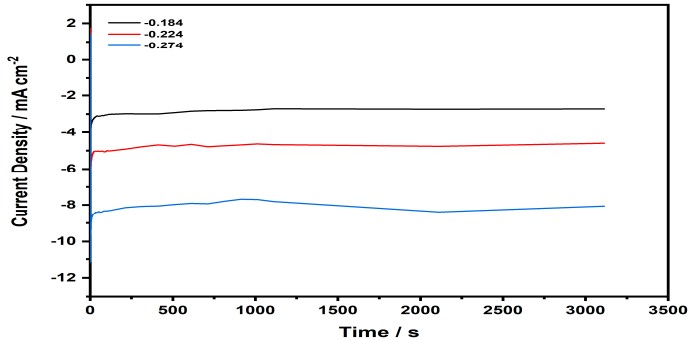

**Figure 12.** Current density (mA·cm$^{-2}$) vs. time (s) curve recorded for the Au-TPTH-P electrode in acidic media (0.1 M H$_2$SO$_4$ and 1 M KNO$_3$); the stepped potentials were −0.184, −0.224, and −0.274 V vs. RHE.

### 2.2.6. Proposed HER Mechanism at the Au-TPTH-P Electrode

Based on the results obtained above from the Tafel slopes, the HER mechanism on the surface of the Au-TPTH-P electrode was deduced. The Tafel slope was 81 mV·dec$^{-1}$, which implies that the reaction proceeded via the Volmer–Heyrovsky mechanism, as previously reported in the literature [23]. Scheme 2 shows the proposed HER mechanism on the Au-TPTH-P electrode. The first (**1** to **4**) steps demonstrate the reduction of the porphyrin scaffold, followed by protonation of the central nitrogen. In the last step (**5**), two hydrogens are electrochemically combined at the central nitrogen, which results in molecular hydrogen generation [28].

**Scheme 2.** The proposed mechanism for hydrogen generation on the surface of the Au-TPTH-P–SAM electrode in acidic medium, adopted and modified from Yanyu Wu et al. [28].

### 3. Materials and Methods

Materials, solvents, and chemicals were purchased from Sigma-Aldrich, Sharjah, UAE, and used directly. Tetrakis (*p*-thioanisole) porphyrin (1) was synthesized as previously reported [17]. Deionized water (18.2 MΩ cm) was used to prepare all solutions. A BRUKER spectrometer (850 MHT), Billerica, MA, USA, was used to measure nuclear magnetic resonance (NMR) spectra and deuterated chloroform (CDCl$_3$) was used as a solvent. An Agilent Technologies Cary 630 FTIR spectrometer, Santa Clara, CA USA, was used to measure the infrared spectra. A Shimadzu (QP-2010 PULS), Kyoto, Japan, mass spectrometer was used to measure the mass spectra. A GENESYS 10 s UV/Vis spectrophotometer (Menlo Park, CA, USA) was used to measure the UV/Vis spectra and dichloromethane (DCM) was used as a solvent. An SP150e (Bi)potentiostat (Bio-Logic SAS), Paris, Frace, was used for all electrochemical measurements and characterizations. The system was configured with an electrochemical impedance module and run on Bio-Logic Ec-Lab software V 11.27, released February 15, 2019. A graphite rod made from Ag/AgCl/KCl (sat.) was used as the counter electrode, with a Luggin probe positioned near the electrode surface. Electrochemical impedance spectroscopy (EIS) data were recorded in the frequency range of 100 KHz to 10 mHz. The sinusoidal signal amplitude was 5 mV from peak to peak at the open circuit potential. The working electrode potential was measured with respect to the reference

electrode and then converted for the reversible hydrogen electrode (RHE) according to Equation (4) [18]:

$$E_{RHE} = E_{Ag/AgCl/KCl(sat.)} + 0.197 \text{ V} + (0.0591 \text{pH}) \tag{4}$$

### 3.1. Synthesis of 5,10,15,20-Tetrakis (p-Thiophenol) Porphyrin (2)

An amount of 200 mg (0.25 mmol) of tetramethylthiol porphyrin (**1**) was dissolved in 10 mL of dry DMF. Zinc chloride ($ZnCl_2$, 200 mg, 1.4 mmol) was added to the solution at room temperature. Then, the reaction mixture was heated in a water bath for 20 h. The resulting dark solution was left to cool to room temperature and then poured into an ice-cold water mixture. The obtained solid was collected via filtration, washed several times with distilled water, and left in an oven (100 °C) to completely dry. The solid product was dissolved in 5 mL dichloromethane (DCM), 2 mL methanol, and 0.5 mL of conc. $H_2SO_4$ was added gradually with continuous stirring. The reaction mixture was stirred at room temperature for eight hours, and then 5 mL of water and 0.5 mL of triethylamine were added. The organic layer was collected, washed several times with water, kept over anhydrous $MgSO_4$, filtered off, and evaporated under vacuum to yield 180 mg (96%) of the free-base of 5,10,15,20-tetra (*p*-thiophenol) porphyrin (TPTH-P) (**2**) (recrystallized from MeOH/DCM 1%).

UV-Vis (DCM): λ = 421, 525, 555, 605, 655 nm. IR: υ = 3300, 1100, 996 cm$^{-1}$ (NH), 2850 cm$^{-1}$ (S-H), 1605–1590 cm$^{-1}$ (Ar). $^1$H-NMR (CDCl$_3$): δ (ppm) = −2.62 (s, 2H, 2NH), 3.82 (s, 1H, S-H), 7.62–8.11 (2 m, 16H, Ar-H), 8.96 (m, 16H, β-H).

### 3.2. SAM Formation on the Gold Electrode

TPTH-P was dissolved in chloroform to produce a 10 mM solution. A gold disk (4 mm diameter) was used as the working electrode, which was polished with alumina powder (0.5 μm diameter) and thoroughly rinsed with water. Then, the clean electrode surface was soaked in thiol-porphyrin solution for 6 h to ensure the chemical bonding of the SAM to the gold electrode. After that, the electrode was washed thoroughly with chloroform to remove the weakly bound thiol-porphyrin from the gold surface, followed by washing with deionized water.

### 3.3. The Contact Angle Measurement

The contact angle measurement was used to investigate the formation of the SAM on the gold surface. In this method, the electrode was held perpendicularly to allow the droplet of water (5 μL) to be placed on the gold surface. Subsequently, the contact angle between the water droplet and the substrate surface was measured.

### 3.4. Electrochemical Measurements

All electrochemical measurements and characterizations were performed as mentioned above.

## 4. Conclusions

The free-base 5,10,15,20-tetrakis (*p*-thiophenol) porphyrin (TPTH-P) (**2**) was synthesized with an excellent purified yield using a new and unique approach. A self-assembled monolayer (SAM) of TPTH-P (**2**) was formed on the gold electrode. SAM formation was confirmed by contact angle measurements and cyclic voltammograms recorded in a 10 mM ferricyanide solution. The catalytic activity of the TPTH-P-modified gold electrode in the HER was investigated in 0.1 M $H_2SO_4$ and 1 M $KNO_3$. Initially, the Au-TPTH-P electrode exhibited poor catalytic activity in the HER. However, after activating the electrode via consecutive cycling processes in 0.1 M $H_2SO_4$ and 1 M $KNO_3$, the catalytic activity was significantly enhanced. The catalytic activity was evaluated by measuring the Tafel slope and overpotential, which were 81 mV·dec$^{-1}$ and 127 mV, respectively. Therefore, based on the outcomes of this work, two factors must be taken into consideration regarding the

investigated thiol-porphyrin SAMs. The first factor is the immersion time; soaking the electrode for 6 h improved the quality of the SAM. The second factor is the activation protocol, which drastically enhanced the organization of the SAM on the gold electrode, which was reflected in its catalytic activity in the HER. Based on these promising outcomes, porphyrin-based SAM electrocatalysts are an amazing class of electrocatalysts due to their versatility and tuneability, opening doors to several applications in energy conversion systems, such as regenerative fuel cells.

**Author Contributions:** Conceptualization, I.E., A.S.A. and M.E.A.; methodology, I.E. and M.E.A.; resources, I.E. and M.E.A.; writing—original draft preparation, I.E., A.S.A. and M.E.A.; writing—review and editing, A.S.A.; visualization, I.E. and M.E.A.; supervision, I.E.; project administration, I.E.; funding acquisition. All authors have read and agreed to the published version of the manuscript.

**Funding:** This research was funded by the Dean of Scientific Research at King Faisal University, grant number 4210.

**Data Availability Statement:** The raw/processed data generated in this work are available upon request from the corresponding author.

**Acknowledgments:** This work was supported by the Deanship of Scientific Research (DSR) at King Faisal University, Vice Presidency for Graduate Studies and Scientific Research, King Faisal University (KFU), Saudi Arabia (Grant number 4210).

**Conflicts of Interest:** The authors declare no conflict of interest. The funder had no role in the design of the study; in the collection, analysis, or interpretation of data; in the writing of the manuscript; or in the decision to publish the results.

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
