# Peer review of "Hydrogen Gas Generation Using Self-Assembled Monolayers (SAMs) of 5,10,15,20-Tetrakis (p-Thiophenol) Porphyrin on a Gold Electrode"

_catalysts, doi:10.3390/catal13101355_

Round 1

Reviewer 1 Report

-

Author Response

Reviewer 1 comments and authors response:

The authors synthesized free base 5, 10, 15, 20-tetrakis (p-thiophenol) porphyrin (TPTH-

  1. P) by a new and unprecedented approach. The self-assembled monolayer (SAM) formation

was confirmed by either contact angle measurements and electrochemically. They also

tested its catalytic activity towards HER. There are only a few problems the authors have to

adjust before the paper can be published. The most significant ones are:

1) The nuclear magnetic map was not processed, and other miscellaneous peaks were

marked in page 4, 5. It is recommended to re-peak and mark the specific position of the

peaks. and mark the name of the substance on the drawing.

Response: 

we thank the reviewer, the integration of all the H-NMR spectra were assigned on both structure and peaks for all porphyrins,  the new figures were replaced in to manuscript.

2) It is suggested to mark the characteristic peak position in the infrared map to clearly

compare the difference between the two substances in page 6.

Response:

 we thank the reviewer, all the IR spectra were repeated for all compounds and the new figures were replaced in to manuscript.

3) In this paper, some writing formats need to be carefully checked and corrected. For

example, 50mV s-1 in line 195 of page 8 needs to be modified, and the garble in the

table of page14 needs to be modified.

Response:

modified and corrected.

4) I can't understand the citation format of the references in page12, 278 [20]

Response:

modified and corrected.

5) In Figure 9, after Au is activated by applying voltage, its surface undergoes some

changes. Therefore, it is necessary to give the Au material sufficient time to activate

before testing. Only after activation can effective comparative results be obtained. This

is because the activation process may cause changes in the properties of the Au

material. Therefore, in order to ensure accuracy, the Au material should be adequately

activated before testing.

Response:

We thank the reviewer for raising this point. As you know the consecutive cycling of gold electrodes in H2SO4 is commonly used to polish the surface and sought to make it very smooth surface. The smooth, uniform, and flat gold surface will improve the assembly of the SAM layer and make more molecules available to enhance the catalytic activity. Applying 10 activation cycles was found to be the optimum conditions to get the best catalytic activity. We have not observed an increase in the activity when applying more than 10 activation cycles.

6) The authors can use the following references to enrich your introduction: Exploration

2021, 1, 20210021. https://doi.org/10.1002/EXP.20210021;

ChemicalEngineeringJournal436,135186,2022.https://doi.org/10.1016/j.cej.2022.13518

6 and Catalysts 2023, 13(9), 1242; https://doi.org/10.3390/catal13091242

Response:

we thank the reviewer for his suggestion, but when we checked the suggested references, we found that, they are irrelevant and using different approach for hydrogen evolution reaction not related to SAM, which we are discussing in the introduction of this manuscript. 

Reviewer 2 Report

In this paper a p-thiophenol porphyrin was synthesized and characterized in order to form a self-assembled monolayer on gold electrode. The authors proved that the modified electrode enhanced the hydrogen evolution reaction.

Moreover, the authors should address the following issues:

1.      At the introduction part Future tense should not be used (lines 82-97).

2.  The authors should add the integration in all 1H-NMR spectra and the deuterated solvent that they conducted the spectra. The integration will help to elucidate if the porphyrin had any impurities present. Also, the protons of each peak should be added at the experimental part (lines 388-390).

3.   The authors should mention the solvent of all UV-Vis spectra at the results and discussion part

4.   The FTIR spectra of tetrakis (p-thiophenol) porphyrin (2) (Black) should be repeated - the peaks from 2000-500 cm-1 are not clear.

5.   The manuscript needs some English editing, especially the introduction and the conclusion parts.

1.       The manuscript needs some English editing, especially the introduction and the conclusion parts.

Author Response

Reviewer 2 comments and authors response

 This paper a p-thiophenol porphyrin was synthesized and characterized in order to form a self-assembled monolayer on gold electrode. The authors proved that the modified electrode enhanced the hydrogen evolution reaction.

Moreover, the authors should address the following issues:

  1. At the introduction part Future tense should not be used (lines 82-97).

Response: done and changed in the manuscript.

  1. The authors should add the integration in all 1H-NMR spectra and the deuterated solvent that they conducted the spectra. The integration will help to elucidate if the porphyrin had any impurities present. Also, the protons of each peak should be added at the experimental part (lines 388-390).

Response:

we thank the reviewer, the solvent was CDCl3 and the deuterated solvent was D2O ,modified in the captions also in the material and method section. he

Additionally, in 1H-NMR spectra, the structures of the porphyrins with the assigned protons and integrations were checked and added to the figures.

3          The authors should mention the solvent of all UV-Vis spectra at the results and discussion part

Response:

we thank the reviewer, the solvent was dichloromethane (DCM) was added and mentioned in the captions..

4- The FTIR spectra of tetrakis (p-thiophenol) porphyrin (2) (Black) should be repeated - the peaks from 2000-500 cm-1 are not clear.

Response:

we thank the reviewer, all the IR spectra were repeated for all compounds with assigning the characteristic signals, and the new figures were replaced in to manuscript.

  • The manuscript needs some English editing, especially the introduction and the conclusion parts.

Response:

we thank the reviewer, the whole manuscript was checked, and the language was revised.

Round 2

Reviewer 1 Report

 The manuscript has been sufficiently improved after revision that warrants itspublication in Catalysts.

Reviewer 2 Report

Accept in current form